# The Nature of Immune Responses to Influenza Vaccination in High-Risk Populations

**DOI:** 10.3390/v13061109

**Published:** 2021-06-09

**Authors:** Kristin B. Wiggins, Maria A. Smith, Stacey Schultz-Cherry

**Affiliations:** Department of Infectious Diseases, St. Jude Children’s Research Hospital, Memphis, TN 308105, USA; Kristin.Wiggins@STJUDE.ORG (K.B.W.); Maria.Smith@STJUDE.ORG (M.A.S.)

**Keywords:** influenza, immunity, vaccine, obesity

## Abstract

The current pandemic has brought a renewed appreciation for the critical importance of vaccines for the promotion of both individual and public health. Influenza vaccines have been our primary tool for infection control to prevent seasonal epidemics and pandemics such as the 2009 H1N1 influenza A virus pandemic. Certain high-risk populations, including the elderly, people with obesity, and individuals with comorbidities such as type 2 diabetes mellitus, are more susceptible to increased disease severity and decreased vaccine efficacy. High-risk populations have unique microenvironments and immune responses that contribute to increased vulnerability for influenza infections. This review focuses on these differences as we investigate the variations in immune responses to influenza vaccination. In order to develop better influenza vaccines, it is critical to understand how to improve responses in our ever-growing high-risk populations.

## 1. Introduction

The ongoing severe acute respiratory syndrome coronavirus 2 (SARS-CoV-2) pandemic signals the growing importance of vaccines for infection control. Effective vaccines must reduce infection rates, disease severity, or transmission of a disease [1]. Several vaccines have been developed and tested since the onset of the pandemic, eight of which were shown to be highly effective in clinical trials and three of which are now authorized for emergency use in the United States [2]. However, vaccine hesitancy continues to threaten the likelihood of safely achieving herd immunity [3], which is needed to protect high-risk populations from coronavirus disease 2019 (COVID-19), including pregnant women, children, the elderly, the immunocompromised, and individuals affected by malnourishment, obesity, and/or chronic pulmonary, metabolic, or cardiac disease [4,5,6,7].

Similar to SARS-CoV-2, influenza viruses also pose a threat to global health with serious annual epidemics [8]. The 2019 influenza season resulted in 38 million cases, which accounted for over 10 billion dollars in direct medical expenses solely in the United States [9,10]. Many of the same high-risk populations who are susceptible to COVID-19 are equally susceptible to contracting influenza virus infections [8]. These individuals possess altered immune systems that contribute to changes in both influenza pathogenesis and viral evolution [11,12]. Notably, unique host factors also play a role in vaccination response.

Vaccination is our main line of defense in controlling influenza epidemics and pandemics. Unfortunately, current standard-of-care vaccines are less effective in protecting high-risk populations, particularly those struggling with obesity. Despite seroconversion rates in obese individuals being similar to those of their healthy weight counterparts after vaccination, obese individuals are nearly twice as likely to contract the influenza virus and experience more severe symptoms [13]. Although obese adults are also more likely to be vaccinated for influenza than non-obese adults [14], obese adults typically experience poor initial and adaptive immune responses to vaccination, which compromise their long-term protection [4]. Even adjuvant-containing vaccines, which elicit more neutralizing and non-neutralizing antibodies than those without, are ineffective in protecting obese mice from homologous influenza challenges [15,16].

When addressing the obesity epidemic, body mass index (BMI) is frequently used as a screening metric to define underweight, healthy weight, overweight, and obesity categories. Worldwide, an estimated two billion people are considered to be overweight or obese, and although obesity accounts for only one-third of that population, over half of the morbidity and mortality rates are contributed by individuals within this subgroup [17]. Because half of the American population is projected to be obese within the next decade [18], the lack of vaccine efficacy from our standard-of-care regimens poses serious public health risks.

Obesity is associated with heart disease, type 2 diabetes (T2DM), chronic kidney disease, and a variety of cancers [17]. One surveillance study found that T2DM tripled an individual’s risk of hospitalization due to influenza virus infection and quadrupled their risk of admission to an intensive care unit once admitted [19]. To investigate fluctuating glucose levels, which are characteristic of T2DM during influenza infections, either constant- or variable-release glucose pumps were inserted into prediabetic diet-induced obese (DIO) mice [20]. Upon influenza challenge, mice with glycemic variability lost more weight and had increased disease severity than did those with constant glucose levels [20]. This suggests that the inconsistent glucose levels characteristic of T2DM directly influence influenza pathogenesis. Unfortunately, few studies have analyzed the effects of T2DM on influenza vaccine efficacy [21]. The growing prevalence of T2DM worldwide [22] is disconcerting because T2DM has moved from the eighteenth to the ninth leading cause of mortality within the last century [23].

Elderly individuals comprise another ever-expanding high-risk population for influenza complications and mortality. One meta-analysis found that elderly individuals who received a high-dose influenza vaccine were approximately 25% less likely to contract the disease and had higher seroprotection rates than did individuals who received a low-dose vaccine [24]. Another study reported that adjuvant-containing vaccines in community settings were more effective than were non-adjuvant vaccines. The adjuvant-containing vaccines protected elderly individuals against symptomatic infections and hospitalizations caused by severe disease, acute coronary syndrome, or cerebrovascular accidents [25]. Despite increased rates of vaccination over the past several decades, influenza-related mortality has not decreased in elderly people [26]. This is particularly concerning because the elderly population accounts for 90% of all influenza-related deaths [26]. Because elderly individuals typically experience vaccine-induced immunogenicity of 30% to 40% [27], they may benefit from a vaccine capable of generating more broadly reactive, long-term responses to better protect them upon infection.

Understanding the critical immunologic responses within high-risk populations will better inform the development of annual seasonal influenza vaccines and result in more effective protection for high-risk populations. Here, we discuss how the microenvironments of high-risk populations affect host immune responses after influenza vaccination and the significant overlap between them.

## 2. Influenza Vaccine Overview

### Standard-of-Care Vaccines

Current influenza vaccines include inactivated, recombinant, and live-attenuated influenza vaccines (LAIVs). The Centers for Disease Control and Prevention do not express a preference for any vaccine platform [28]. Inactivated egg-based and cell-based vaccines are recommended for individuals over the age of 6 months and 4 years, respectively [28]. Recombinant vaccines are approved for individuals 18 years and older [28]. LAIVs are approved for people between the ages of 2 and 49 years who are not considered part of a high-risk population [28]. For individuals aged 65 years and older, high-dose and adjuvant-containing vaccines are available to facilitate stronger immune responses after vaccination [29].

Inactivated vaccines can be manufactured in one of the following three compositions: (1) subunit vaccines containing surface glycoproteins, such as hemagglutinin (HA) and neuraminidase (NA); (2) split-virus vaccines containing viral proteins without the lipid envelope; or (3) whole-virus vaccines containing chemically inactivated and purified whole virus [30]. To protect against all four strains of seasonal influenza viruses, recombinant vaccines contain synthetic HA subunits, whereas LAIVs contain whole attenuated viruses [30]. These vaccines are manufactured as either trivalent or quadrivalent formulations, the latter being more frequently used for increased protection. Trivalent vaccines contain the influenza A subtypes H1N1 and H3N2, along with the predicted dominant influenza B lineage of either Yamagata or Victoria. Quadrivalent vaccines, in contrast, contain both influenza B lineages [29].

The effectiveness of annual influenza vaccines varies between 10% and 60% [10] due in part to antigenic drift. Antigenic drift occurs when mutations lead to changes in the antibody-binding sites in the HA or NA surface proteins, causing vaccine escape [31]. Increasingly important changes may occur in cases of antigenic shift when a completely new HA or NA is introduced, thereby creating a novel influenza subtype. This may occur as a result of viral recombination with animal-derived strains [31]. The trivalent, quadrivalent, and recombinant inactivated influenza vaccines primarily stimulate production of antibodies against epitopes on HA [32]. Vaccines that promote the generation of antibodies targeting the HA head of the virus are highly prone to decreased efficacy because of the frequency of antigenic drift [31]. Because many vaccines offer strain-specific protection, vaccinated individuals are unable to mount the proper antibody response once infected [33].

## 3. Responses to Influenza Vaccination

### 3.1. Humoral Responses

Influenza vaccines are designed to mimic a natural infection to stimulate a protective, long-lasting immune response to future infections. After detection of a vaccine antigen, the innate immune system responds immediately. This initial response is followed by the inductive, effector, and memory phases of the immune response, each coordinated by the adaptive immune system. Humoral immunity encompasses all of the antibody and B cell-mediated responses. This is a rapid process upon detection of an antigen in which B cells assist with clearance and protection via sterilizing or broadly neutralizing antibodies to prevent further infection. Proper B-cell activation is critical for effective immune responses to vaccines. Furthermore, B-cell fate determination is driven by infection-induced innate signals to drive early extrafollicular responses and expansion. If these signals are identified, vaccine designs can be tailored to target them for initiating rapid immune protection [34]. B-cell diversity and subset distribution are two important characteristics in healthy immune systems that comprise a diverse B-cell receptor repertoire, broadly reactive HA and NA neutralizing antibodies, and healthy proportions of naïve B cells, memory B cells, and plasma cells [35] (Table 1).

Naïve B cells with antigen-specific molecules travel through the lymphatic system and when they interact with an antigen begin the process of differentiation to generate both memory B cells and effector cells [36]. B cells produce antibodies with shared antigen-binding sites, which enable them to locate their targets by using one of three mechanisms [36]. The first mechanism of neutralization is a process during which antibodies bind to pathogens, preventing them from entering any of the host cells [36]. The second mechanism of opsonization involves pathogen-attached antibodies binding to phagocytes via Fc receptors for uptake [36]. The third mechanism of complement activation occurs when antibodies bind to the pathogen’s surface and activate the classical complement pathway to recruit more immune cells to the site of the infection by increasing opsonization rates [36]. Standard seasonal influenza vaccines drive antibodies toward the neutralization mechanism targeting the viral HA head.

Class switch recombination occurs when immature B cells expressing IgM in the germinal center switch to one of the other immunoglobulin classes before infection. T follicular helper (T_FH_) cell cytokine presence dictates the expanded isotype, specifically IgM, IgD, IgG, IgE, or IgA. Each isotype contributes different functions in the immune system. Notably, class switching from IgM to IgG and vaccine-specific IgG production are impaired in obese patients, although no studies have determined the mechanism for this [37,38] (Table 1). The Fc-Fc receptor plays a critical role in affinity maturation via interactions with IgG immune complexes [39,40]. Driving the maturation of B cells with particular Fc receptor-binding properties to create CD23-elicited antibodies, which are more broadly protective and have higher affinity than HA-specific antibodies, is proposed for universal influenza vaccine platforms [41]. Enhanced interactions with altered glycosylation profiles occur in the context of HIV antibodies, which are broadly neutralizing with higher titers and improved B-cell responses via recruitment of complement [42]. This strategy may enhance vaccine efficacy in obese patients. However, prior vaccination history may lead to decreased antibody affinity maturation to HA domains, resulting in decreased vaccine efficacy [43].

### 3.2. Cell-Mediated Responses

Cell-mediated immunity embodies all of the T cell-mediated responses, including helper T cells (T_H_), killer T cells, and macrophages. This is a slower process than are humoral responses in which T cells are continuously circulating in the blood and lymphatics to identify antigen-presenting cells with an antigen [36]. The maturation and differentiation of naïve T cells into T_H_ or killer T cells depends on antigen-presenting cells with MHC class I proteins for binding specificity [36]. When T_H_ cells interact with antigen-presenting cells, T_H_ cells can release cytokines to activate killer T cells and send macrophages to the sites of foreign particles [36].

T-cell diversity and function are two important characteristics of healthy immune systems, containing diverse T-cell receptor repertoires; proper levels of activation, differentiation, and cytokine production; and healthy proportions of naïve T cells, effector T cells, and memory T cells [35]. The incredible diversity of the T-cell receptor repertoire encompasses its abundance, functionality, and specificity, which poses considerable challenges for eliciting vaccine-conferred T cell-mediated immunity to influenza infections [44].

Once activated, a CD8^+^ T cell will go through a period of expansion, and once the pathogen is controlled, the CD8 T cell population will contract via apoptosis and establish memory cells that circulate the blood and lymph [45]. Robust CD4^+^ and CD8^+^ T-cell responses to influenza infections can protect patients from severe disease, but these responses must be induced in a carefully balanced way to prevent excessive proinflammatory reactions and increased immunopathology [45].

The broad specificity of CD4^+^ T cells results from their heterozygosity at HLA loci, but most circulating cells are reactive to the conserved influenza matrix protein [44]. HLA class II molecules can present as different isotypes, including HLA-DR, HLA-DQ, and HLA-DP [44]. These responses can be enhanced by CD4^+^ T cells, which contribute to better priming, expansion, and longer-lasting memory. Enhanced stimulation of the expansion of CD4^+^ T cells, which is critical for protecting the lungs from viral replication along with triggering robust antibody responses, results in higher levels of cross-reactive CD4^+^ T cells with a greater ability to recognize novel influenza strains [44]. Vaccine strategies that leverage the effector functions of CD4^+^ T cells have been proposed as a potential step towards universal influenza vaccines [44]. The high mutation rates characteristic of the influenza virus pose the risk of vaccine escape, so optimizing the memory CD8^+^ T-cell responses when exposed to a novel strain is another proposed vaccine strategy. Unfortunately, cross-reactive clones can also lead to immunodominance which restricts the TCR repertoire and leads to influenza variant escape, the inability to control viral spread, and worsened immunopathology [45].

## 4. Unique Immune Environments in High-Risk Populations

### 4.1. Metabolic Syndrome

Because individuals in the same BMI classification can have a diverse array of different metabolic features based on diets, exercise regimens, and other comorbidities, the BMI metric is not a proper indicator of overall health [75,76]. Revised definitions of obesity are proposed to include the etiology (e.g., drug-induced weight gain, endocrine disease, or major depressive disorder), degree of adiposity (class I–VI), and overall health risk (low, medium, or high) of patients [77]. Recently, labels indicating metabolic health were also proposed for each BMI category to account for potential conflicts between body status and internal metabolism. For example, metabolically healthy obese individuals possess healthy metabolic markers, despite increased adiposity, whereas metabolically unhealthy individuals in the healthy weight category possess abnormal metabolic markers, despite decreased adiposity [75]. Unhealthy metabolic markers include high fasting glucose levels or insulin resistance; high cholesterol, triglyceride, or lipid levels; and elevated blood pressure [78]. According to one meta-analysis [75], metabolic health may be a stronger predictor of overall health than weight or BMI alone. Metabolically unhealthy individuals in all BMI categories had an elevated risk of mortality compared to their metabolically healthy counterparts [75].

The four major intertwined hallmarks of metabolic syndrome (MetS) include visceral adiposity, insulin resistance, dyslipidemia, and endothelial dysfunction [78]. MetS is useful for predicting which patients are at increased risk of cardiovascular disease or T2DM [78]. Additionally, MetS may have important implications for host susceptibility and immune responses to viral infections and vaccine efficacy [67,79,80]. Glucose-intolerant rats exhibit decreased immunologic memory [81], and patients with T2DM who receive hemodialysis experience elevated risk of vaccine failure after hepatitis B vaccination [82]. Overweight children and adults are at increased risk of tetanus [83] and influenza [84] vaccine failure as well. Obesity and MetS considerably affect vaccine response because of an increased risk for inherent chronic inflammation and metabolic tissue stress that alter leukocyte activity [85].

### 4.2. Adipose Tissue

Adipose tissue (AT) is distributed throughout the body as subcutaneous or visceral fat and is categorized into two major types: white AT, which stores energy in the form of triglycerides, and brown AT, which dissipates energy through the production of body heat [86]. In addition, beige adipocytes are now considered a third type of AT. Beige adipocytes are distributed throughout white AT and are more resistant to inflammation, which plays a protective role against obesity-related disorders [87]. Interestingly, increased body weight appears to be associated with reduced brown AT in age and obesity-related metabolic disorders [86]. AT is composed of many different cell types, such as adipocytes and their progenitors, endothelial cells, and immune cells [88].

Obesity is marked by substantial dysregulation in the physiologic responses stemming from the constant state of low-grade inflammation caused by constitutive activation of the innate immune system. Increased body weight results in the expansion of AT, which increases host proinflammatory signals and dysregulates homeostatic immune signals [68]. A study by Li et al. observed that deficiencies in murine maternal vitamin D levels often lead to obesity in their progeny and create polarization within the adipose deposits in the epididymal and inguinal white AT [89]. This polarization affects cytokine expression and immune profiles, highlighting the importance of complex interactions with the AT in dictating immune responses. Vitamin A represents another interesting interaction between AT and nutrients. Retinoic acid, the transcriptionally active form of vitamin A, is linked to obesity and metabolic disease [90]. Adipocytes are critical for the metabolism of retinoic acid, which can modulate AT inflammation and adipocyte differentiation [90]. Penkert et al. found that vitamin A supplementation in DIO mice leads to decreased systemic inflammation, increased antibody responses after vaccination, and decreased viral titers and overall reduced disease severity after viral challenge [91]. Nutrition is an extensively studied factor that plays an important role in immune responses. AT is a highly dynamic, metabolically active tissue that responds to changes in nutrition and is capable of secreting lipids, metabolites, proteins, extracellular vesicles, hormones, and non-coding RNAs that allow for communication and influence on immune responses [92].

The specific resident immune cells within the AT of obese patients reveal unique cellular and molecular interactions in the inflamed microenvironment. Macrophages that express TNF-α, iNOS, and IL-6 accumulate in the AT, which contribute to inflammation [93] (Table 1, Figure 1). AT inflammation is a critical event leading to MetS, insulin resistance, and other obesity-associated diseases [64,68]. The macrophages in obese patients can enter the AT to promote insulin resistance and T2DM by converting anti-inflammatory M2 macrophage alternative states to proinflammatory M1 macrophage classical states [46] (Table 1, Figure 1). The accumulation of AT macrophages is associated with increased body weight and insulin resistance [47]. Lean AT is primarily composed of adipocytes, high rates of M2 macrophages, and low rates of M1 macrophages [46] (Table 1). In contrast, the AT of patients who are obese or have MetS is composed of hypertrophic and necrotic adipocytes, low rates of M2 macrophages, and high rates of M1 macrophages [46] (Table 1). Diabetic AT is even more complex, with increased rates of M1 macrophages, necrotic adipocytes, and signs of fibrosis [46] (Table 1). M1 polarization in AT also occurs in elderly patients [48] (Table 1). Macrophages are able to modulate insulin sensitivity and adipogenesis, emphasizing their critical role in high-risk populations [94]. Decreased influenza vaccine efficacy in obese patients may be partially attributed to suboptimal macrophage activity and maturation within the AT [4].

### 4.3. B Cells

B cells serve a fundamental purpose for humoral immunity by regulating inflammation within the AT. However, obesity with or without MetS reveals characteristic B-cell defects that cause inflammation and autoimmune antibody release. This occurs in elderly individuals, as well [56] (Table 1, Figure 1). Indeed, the term “adipaging” was coined to highlight how the chronic inflammation and dysfunctional AT in obese patients exhibits similar effects as aging [57] (Table 1, Figure 1). Accelerated aging related to obesity is intricately associated with abnormal AT function, which may explain decreased vaccine responses in these individuals [57]. Upon natural influenza infection or vaccination, both obese and elderly populations experience decreased memory B-cell responses against novel antigenic stimulation [58] (Table 1). Late memory B cells accumulate at high levels in the blood of both elderly and obese individuals. These cells are not responsive to influenza infection and secrete antibodies with only autoimmune specificity, resulting from their constitutive activation [59] (Table 1). Elderly individuals also face severely restricted B-cell receptor repertoires, expression of highly specific, non-neutralizing antibodies, and higher levels of age-associated B cells and atypical memory B cells [35] (Table 1). Vaccination does not improve these immune responses. Compromised vaccine antibody responses also occur with less class switching in B cells and greater counts of proinflammatory-exhausted memory B cells, which in turn reduces the longevity of seroprotective responses [60] (Table 1). Overall, immunosenescence leaves patients susceptible to severe infections as a result of their deteriorating immune systems.

Immunologic imprinting also affects host immune responses to vaccination. The influenza strains encountered during childhood may confer lasting effects on immune memory. Encounters with influenza strains early in life may prevent the immune system from stimulating new responses to current strains [61]. Individuals struggle to mount equally strong responses to novel strains they encounter as adults than to the strains they encountered during childhood [61]. Immunodominance restricts the immune system from mounting broadly protective memory B-cell responses, which are maintained for prolonged periods of time [61] (Table 1, Figure 1). This phenomenon was observed in survivors of the Spanish influenza pandemic. B cells secreting specific binding antibodies for that particular influenza virus strain did not cross react with contemporary strains 90 years later [95]. The issue of immunologic imprinting can severely compromise the efficacy of vaccinations over the lifespan of an individual [96]. This suggests that universal influenza vaccines should be targeted for children in an effort to prime their immune responses and curb them to be more broadly reactive and less strain specific. Understanding the core mechanisms regarding an individual’s immune history, and especially how it varies in high-risk populations, may be critical for influenza vaccine development [97].

High-risk populations experience altered B-cell responses, whereas these responses are maintained in their healthy counterparts. Zhai et al. compared B-cell activation rates between nondiabetic, diabetic obese, and lean healthy subjects to understand the role B cells play in diabetes pathogenesis [62]. Phenotypically, the B cells of obese individuals exhibit heightened levels of activation, leading to markedly increased differentiation in antibody-producing plasmablasts [62] (Table 1). All of the obese subjects in their study experienced higher levels of IL-6 and TNF-α secretion from B cells that promotes elevated secretion of proinflammatory cytokines than those of their healthy, lean counterparts [62] (Table 1). Interestingly, diabetic obese subjects experience deficient IL-10 secretion, which plays a critical role in mediating immune suppression by regulatory B cells [62] (Table 1). Furthermore, subjects who are diabetic and obese have higher B-cell activation levels but impaired responses to novel antigen stimulation after influenza vaccination, despite having similar inflammatory profiles with those obese non-diabetic individuals [62] (Table 1). The interaction between antigen-specific B cells with T_FH_ cells in obese patients may be compromised and skew the differentiation, proliferation, and survival of first-line defense cells in diabetic individuals [62].

Additionally, intrinsic B-cell differences affect vaccine responses. Diaz et al. analyzed the effects of metformin on influenza vaccine responses in both obese and diabetic patients. Clinicians frequently prescribe metformin for T2DM, which has wide-ranging effects on cellular processes via the regulation of AMP-activated protein kinase and mammalian target of rapamycin [98]. The activation of AMP-activated protein kinase is favorable in obese patients because it is largely dysregulated and expressed at low levels, which is a proposed cause of MetS [99]. Metformin lowers B cell-intrinsic inflammation and increases antibody generation to heighten the responses to influenza vaccines in vitro and in vivo [98]. Therefore, metformin drug may be a promising tool for treating inflammatory-based and age-related conditions.

The outcome of proper B-cell activation and activity is the creation of functional antibodies. Karlsson et al. reported that although obese mice exhibit similar neutralizing and non-neutralizing antibody titers and seroconversion rates as that of lean mice after vaccination, obese mice are not protected during viral challenge. Indeed, vaccinated obese mice experience increased viral loads and severely delayed viral clearance [15]. However, the breadth and magnitude of humoral responses with respect to HA and NA is markedly reduced in DIO mice [15]. Clearly, antibodies are not a sufficient measure of influenza vaccination-conferred protection. Further research is warranted to investigate these vaccine-elicited humoral immune responses in patients at high risk of severe influenza infections.

### 4.4. T Cells

Cell-mediated immunity is relied upon for clearance of infections in individuals with impaired humoral immunity and decreased vaccine efficacy, emphasizing the importance of this immunologic cascade in patients at high risk of severe influenza infections. Unique inflammatory dendritic cells are generated from inflammatory monocytes to help regulate AT inflammation, accentuating the role of inflammatory dendritic cells in obesity and MetS [64]. Mice placed on a high-fat diet exhibit increased numbers of inflammatory dendritic cells that induce the differentiation of proinflammatory CD4^+^ T_H_17 cells, in contrast with T_H_1 cells, while polarizing M1 cells [64]. T_H_17 cells often cause the inflammation associated with influenza virus infections and can trigger autoimmune disease, but they can also help control certain bacterial infections. T_H_1 cells are typically required for increased resistance to influenza virus infections. Both cell types are induced in parallel and are balanced in healthy patients [49] (Table 1). However, in obese patients, the accumulation of macrophages in the AT occurs in a CC chemokine receptor 2-dependent manner and in an independent manner in dendritic cells [50]. This suggests that the inflammatory conditions characteristic of obese AT microenvironments lead to the increased macrophage and monocyte accumulation that is typical in diseased tissues affected by autoimmune diseases [51].

The CD11c^+^ dendritic cells in the AT are predicted to play a key role in controlling adaptive immune responses, with antigen presentation and differentiation of naïve CD4^+^ T cells into T_H_17 cells because of CX3CR1 expression. The number of CD11c^+^ cells is also higher in subcutaneous AT than in omental tissue, which may have a stronger effect on the circulating lymph and blood supply [50] (Table 1). Overall, increased T_H_17 cells are harmful in host responses to influenza infections because they lead to increased inflammation, decreased viral clearance, and increased disease severity [52]. T2DM decreased the percentage of CD11c^+^ dendritic cells in subcutaneous adipose tissue, but increased the percentage of CD123^+^ plasmacytoid dendritic cells; such changes may contribute to the dysregulation of the lymphocyte immune response [53] (Table 1). Interestingly, elderly individuals had comparable dendritic cell profiles to their younger counterparts [54] (Table 1). However, the dendritic cells were unable to consistently up-regulate chemokines in response to pathogens, revealing an inability to dependably induce an inflammatory response [54] (Table 1). The extraordinary differences within the ATs and dendritic cells of high-risk patients provide some understanding of decreased influenza vaccine efficacy.

Cellular immune responses to vaccination in obese patients are severely affected by the excessively inflamed microenvironment, which leads to reduced activation, maintenance, and activity of memory T cells [13] (Table 1, Figure 1). Park et al. found considerably decreased influenza virus-specific effector memory CD8^+^ T cells in DIO mice after vaccination. They also observed heightened levels of the inflammatory markers MCP-1 and IFN-γ, as well as the M1 macrophage marker CD64, after viral challenge [16] (Table 1). Furthermore, Paich et al. observed compromised activation of CD4^+^ and CD8^+^ T cells from overweight and obese individuals, which is evidenced by decreased expression levels of CD69 [55] (Table 1). CD28 expression levels are also decreased, thereby lowering costimulatory signaling, expansion, and survival [55] (Table 1). Additionally, the elderly population may experience shifts toward higher expression of anti-inflammatory interleukin-10 (IL-10), which also decreases CD8^+^ T cells [63] (Table 1). Elderly individuals also experience severely restricted T-cell receptor repertoires, higher expression of restricted T-cell clones, and poor activation, differentiation, and cytokine production levels [35] (Table 1, Figure 1). Both elderly and obese populations, as well as those with T2DM, experience immunosenescence and accelerated aging of the immune system, most severely in the CD4^+^ and CD8^+^ T cells [65,66] (Table 1, Figure 1). T-cell dysfunction is also implicated as a major contributor of disease pathology in people with T2DM [66] (Table 1). Increased levels of differentiated end-stage memory T cells mark the cellular senescence that is characteristic of several high-risk populations, contributing to markedly reduced vaccine efficacy.

### 4.5. Important Metabolic Markers

There are several metabolic markers correlating with the presence of MetS, rather than age, that contribute to the systemic inflammation and oxidative stress with repercussions on the immune system [69]. The ATs of obese patients produce high levels of leptin. Increased leptin levels lead to higher food intake and decreased expenditure of energy, while contributing to insulin resistance, T2DM, and cardiovascular diseases [70] (Table 1). Leptin is an adipocyte-derived hormone and a member of the IL-6 superfamily. Leptin is an important starvation signal essential for food intake, energy preservation, and basal metabolism [71,100]. Leptin also regulates both monocyte/macrophage function and proinflammatory responses because it activates costimulated T cells to induce T_H_1 responses [72]. Membrane-bound leptin receptor is a type I transmembrane glycoprotein found on the cell surface of many immune cells, including natural killer cells, macrophages, dendritic cells, regulatory T cells, and B cells [100]. This receptor is important in the JAK-STAT, IRS-1-PI3K, and MAPK signaling pathways, highlighting its broad immunoregulatory functions as a link between the immune system and nutrition [72]. Leptin resistance may occur in obese patients when excessive leptin levels result in attenuated signaling (Figure 1). This is associated with obesity and diabetes, with broad implications on host health [71] (Table 1), including impaired innate and adaptive influenza-specific immunity and consequentially decreased vaccine efficacy [73]. Leptin resistance prevents proper activation of monocytes, macrophages, neutrophils, and natural killer cells, thereby reducing innate immunity (Table 1). Leptin resistance also prevents CD4^+^ T cells from proliferating, expressing adhesion molecules, and secreting IL-2, thereby diminishing adaptive immunity [101].

Leptin-related gene polymorphisms are attributed to differences in antibody production and B-cell responses in humans [102]. Leptin levels may directly compromise B-cell responses in obese patients, leading to decreased vaccine efficacy. A study by Frasca et al. analyzed the effects of leptin on immunosenescence in the B cells of lean and obese individuals [38]. Cell proliferation, as measured by ^3^H-thymidine incorporation assays, is higher in lean individuals [38]. Treating B cells with a physiologically relevant dose of leptin in vitro and then stimulating them with CpG decreases vaccine-specific IgG antibody levels, along with mRNA expression of E47 and AID (both of which are indicative of class switch recombination and B-cell activity) to similar levels as those of obese and elderly individuals [38]. In contrast, IgM levels remain the same, and proinflammatory cytokines, such as TNF-α and other inflammatory markers, are also upregulated after leptin administration [38].

Epigenetic alterations acquired throughout an individual’s life can also yield major immunomodulating effects that hinder the ability of B cells to produce antibodies. This results in exhausted B cells that replace naïve B cells, beginning a cascade of cell-mediated immunity decline that leaves patients with severely compromised immune systems [58]. Leptin may contribute to B-cell immunosenescence through its proinflammatory properties, which may help to explain why people with obesity have generally poor immune responses to influenza infection and/or vaccination [38]. Additionally, epigenetic modifications leading to B-cell immunosenescence have been seen in aging and T2DM [103]. Decreased levels of microRNAs within the AT that normally regulate inflammation have been linked to vascular disease and aging [103]. The characteristic impaired insulin signaling seen can be improved by silencing endothelial NF-KB, a transcription factor that increases the expression of inflammatory genes as a result of the downregulation of sirtuins that serve as cardiovascular aging and inflammation regulators [103]. Overly active NF-KB can therefore alter innate responses and inflammation within the endothelium, providing a potential link between metabolic disease and premature aging associated with obesity and T2DM [103] (Figure 1).

Reduced levels of adiponectin are common in the ATs of obese patients and contribute to glycemia and atherosclerosis [70] (Table 1). Increased proinflammatory cytokines, such as leptin, coupled with reduced anti-inflammatory cytokines, such as adiponectin, may lead to organ-specific diseases and contribute to the onset of T2DM [64,68] (Table 1, Figure 1). Though the direct effects of adiponectin on pathogen titers has not been investigated, low adiponectin levels, in combination with high leptin levels, may contribute to the pathogenesis of MetS and the increased difficulty of clearing infections. Adiponectin diminishes proinflammatory cytokine generation and deters macrophage activation [67] (Table 1). Adiponectin directly controls expression of interleukin 1 receptor-associated kinase M, which modulates macrophage reactions to pathogens and leads to hyperresponsiveness, increased proinflammatory mediators, potential cytokine storms [104], and thereby compromised vaccine responses, when adiponectin levels are low. Overexpression of anti-inflammatory adiponectin in transgenic genetically obese mice improves their metabolic health by normalizing glucose, insulin, and triglyceride levels, despite markedly increased subcutaneous fat mass [74] (Table 1). Additionally, low adiponectin levels occur in pregnant women and individuals with MetS [105], suggesting that low adiponectin levels correlate with increased susceptibility and advanced disease severity in multiple high-risk populations.

## 5. Conclusions

Several intrinsic and extrinsic host factors affect influenza vaccine responses, including immunodeficiencies, comorbidities, immunosenescence due to advanced age, suboptimal health status due to overnutrition or undernutrition, and sex biases [106]. A study by Kuo et al. uncovered sex-specific effects of BMI on influenza vaccine responses among a cohort of health care workers at Johns Hopkins Hospital [107]. The study indicated that neutralizing antibody responses to H3N2 decline with increasing BMI in female patients, suggesting that BMI more strongly affects female immune responses than those of men, which is most likely due to hormonal differences [107]. Therefore, such sex-specific differences should be further investigated and potentially considered in future vaccine design. Effective doses and formulations should be identified for each major host factor to increase vaccine efficacy at a population level.

Obesity can affect vaccine safety in addition to efficacy. People may experience the following adverse reactions to a vaccine: immune-mediated reactions, such as inflammation or redness at the site of injection; multisystem reactions, including anaphylaxis or fever; and/or organ-specific reactions, such as rashes or thrombocytopenia [106]. One literature review described the variation of vaccine safety based on administration route and needle length and found that localized reactions are lower with intramuscular injections than with subcutaneous injections [108]. Longer needles more easily reach the muscle and correspond with fewer instances of edema and erythema, whereas shorter needles often result in subcutaneous injections when increased amounts of subcutaneous tissue are present in overweight and obese individuals [108]. The correct needle length should be selected according to an individual’s body status to avoid causing unnecessary adverse events after vaccination. Personalized medicine, which allows for the adjustment of adjuvants, dosages, formulations, and the routes of appropriate administration according to factors such as sex, age, adiposity, and metabolic health, may minimize unfavorable reactions and maximize efficacy [109].

To overcome the issues of immunosenescence, reduced class switching, and immune imprinting and to avoid the annual revision of vaccine strains, more broadly protective universal vaccines must be created that safely offer sustained protection for people from any population with high effectiveness. Many strides have been made to develop novel universal influenza vaccines across diverse platforms, including virus-like particles, synthetic-viruses, nucleic acids, and viral vectors [30]. Recent studies tout the promise of HA stalk [110], NA [111], or computationally optimized broadly reactive antigen methodology-based [112] vaccines, which appear to offer lasting protection against antigenic drift in mouse models. Another potential avenue is an mRNA-based vaccine, which has been utilized by several companies as a SARS-CoV-2 vaccine platform [2]. The high efficacy rates can be attributed in part to its reactogenic nature. Unfortunately, the increased efficacy comes at the cost of worsened side effects following immunization, leading to vaccine hesitancy [3]. It may not be feasible to implement a highly efficacious and reactogenic annual influenza vaccine if people experience similarly undesired side effects. We must also establish better immune correlates of protection than antibody titers, as determined by HA inhibition assays, because these do not predict influenza protection in high-risk populations. As these novel platforms become validated in animal models of high-risk populations, we can begin to move towards influenza vaccines that are tailored to individual needs, especially because a one-size-fits-all approach has been ineffective to date.

During the last influenza season, the CDC estimates that the number of cases were significantly lower than expected. This is due to several lifestyle changes that were introduced since the onset of the SARS-CoV-2 pandemic. A massive shift toward better hygiene, restricted travel both domestically and internationally, virtual classes, work-from-home, mask requirements, and social distancing not only worked to control the spread of SARS-CoV-2, but also that of influenza. With less influenza circulation and lower rates of influenza vaccination, our immune responses at a population level may not be protected against the strains that are currently circulating. There might be an increase in influenza severity during the upcoming influenza season due to this lack of exposure. Encouraging people to get their annual influenza vaccines as soon as possible will help mitigate these potential issues in the coming year.

Humoral and cell-mediated host immune responses play a critical role in vaccine efficacy beyond just influenza research. The implications of decreased influenza vaccine efficacy in high-risk populations due to altered immune responses must be considered in light of the current SARS-CoV-2 pandemic. Unfortunately, dysregulated immune systems and unhealthy metabolic markers in these populations may lead to compromised vaccine-conferred protection against circulating strains within the community. This warrants immediate further studies to translate what is known about responses to influenza vaccines into the realm of COVID-19 vaccine platforms because the same considerations and limitations apply. We must find alternative ways to increase vaccine efficacy in high-risk populations to better protect them from vaccine-preventable diseases to control epidemics and pandemics and contribute to overall improved global health.

## Figures and Tables

**Figure 1 viruses-13-01109-f001:**
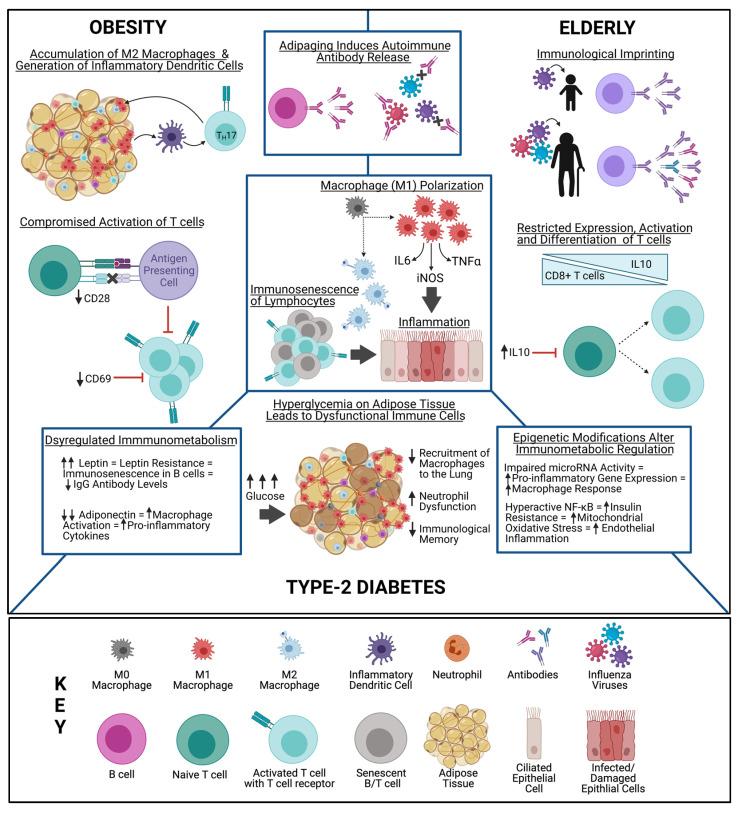
There is a significant overlap between immune responses observed in people with obesity, type 2 diabetes mellitus, or advanced age. The unique immune environment significantly impacts influenza vaccine efficacy among these high-risk populations. Created with Biorender.com.

**Table 1 viruses-13-01109-t001:** Comparison of the immune responses to influenza vaccination in healthy vs high-risk populations. These differences correspond to changes in influenza vaccine efficacy.

	Healthy	Obesity	Type-2 Diabetes	Elderly	References
**Adipose Tissue Composition**	Functional adipocytes; high rates of M2 macrophages; low rates of M1 macrophages	Dysfunctional hypertrophic and necrotic adipocytes; low rates of M2 macrophages; high rates of M1 macrophages	Dysfunctional necrotic adipocytes; increased rates of M1 macrophages; evidence of fibrosis	Dysfunctional necrotic adipocytes; prevalence of M1 macrophages	[46,47,48]
**Dendritic Cells (DCs)**	Functional DCs; balanced levels of anti-inflammatory Th1 cells to polarize M2 macrophages; balanced levels of pro-inflammatory Th17 cells to polarize M1 macrophages	High levels of inflammatory CD11c+ DCs that produce more pro-inflammatory Th17 cells to polarize M1 macrophages	Lower levels of inflammatory CD11c+ DCs with higher levels of inflammatory CD123+ plasmacytoid DCs	Healthy DCs but unreliable ability to induce an inflammatory response	[49,50,51,52,53,54,55]
**B Cells**	Regular levels of activation; regular cytokine release; regular class-switching; generation of broad neutralizing antibodies; active responses to novel antigen stimulation	High levels of activation; increased secretion of IL6 and TNFα that contributes to the pro-inflammatory environment; decreased class-switching; non-neutralizing antibodies; atypical late memory B cells accumulate at high levels in blood; impaired responses to novel antigen stimulation	High levels of activation; compromised regulatory B cell activity due to deficient IL-10 secretion; impaired responses to novel antigen stimulation	Decreased class switching; severely restricted B-cell receptor repertoires; highly specific, non-neutralizing antibodies; atypical late memory B cells accumulate at high levels in blood; impaired responses to novel antigen stimulation	[38,56,57,58,59,60,61,62,63]
**T Cells**	Healthy levels of CD4+ and CD8+ T cell activation; broad T cell receptor repertoires; proper costimulatory signaling; proper T cell expansion	Compromised activation of CD4+ and CD8+ T cells; lower levels of costimulatory signaling; decreased T cell expansion	Compromised activation of CD4+ and CD8+ T cells; T cell dysfunction that contributes to disease pathology	Compromised activation of CD4+ and CD8+ T cells; restricted T cell receptor repertoires; restricted T cell clones; poor activation; poor differentiation	[13,16,35,50,55,63,64,65,66]
**Metabolic Markers**	Balanced levels of adiponectin and leptin; proper insulin release	Low levels of adiponectin; high levels of leptin; high levels of insulin; insulin resistance	Low levels of adiponectin; high levels of leptin; high levels of insulin; insulin resistance	Metabolic markers are more dependent upon MetS status than age	[64,67,68,69,70,71,72,73,74]

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
