# Peer review of "The Nature of Immune Responses to Influenza Vaccination in High-Risk Populations"

_viruses, 2021, doi:10.3390/v13061109_

Round 1

Reviewer 1 Report

This is an interesting and relevant review addressing current conflicts between chronic metabolic disorders suffered by an increasing percentage of the population, and the ability to protect these high risk groups from influenza by vaccination.  The authors review the current literature on how metabolic syndrome affects the efficacy of influenza vaccination, with an emphasis on obesity. They discuss the cell mediated and humoral responses to immunization, and changes in the MetS environment and predominant cells in that dampen the immune response to influenza vaccine and pathogen clearance in general.

Broad comments:

The "Cell-Mediated Responses" section reads as a general and light overview of the CMR in general, and more information or descriptions of studies specific to CD4 and CD8 responses during influenza or IAV immunization would improve this section, particularly as a backdrop to when the authors detail CMRs in the obese environment later in the manuscript.

Specific comments:

Line 202 According to one meta-analysis, metabolically healthy, obese individuals have a higher risk of mortality than do healthy weight individuals. Additionally, metabolically unhealthy groups in all BMI categories have a similarly elevated risk as do their metabolically healthy counterparts [46]. This suggests that metabolic health may be a stronger predictor of health than weight or BMI alone.

It would be helpful to reword this section, because the correlations seem to indicate that metabolic health is not a predictor of mortality, which is contrary to the authors’ conclusions

Line 253 The macrophages in obese patients can enter the AT and convert from an anti-inflammatory M2 alternative state to a proinflammatory M1 classical state to promote insulin resistance and T2DM [68]

Word missing between ‘convert’ and ‘from’

Line 276 Indeed, the terms “adipaging” and “inflammaging” were coined to highlight how the chronic inflammation and dysfunctional AT in obese patients exhibits similar effects as aging

Missing a reference for inflamm-aging (Franceschi et al, 2000).

Line 444 Moreover, overly active NF-B can alter innate responses and inflammation within the endothelium, providing a link between metabolic disease and premature aging that could impact premature aging [102]

This sentence needs to be made clearer or more in detail. Many factors activate NFkB altering inflammation, most of which are not related to premature aging. If the authors could explain in more detail why NFkB activation in this circumstance links metabolic disease and premature aging, that would strengthen the conclusion.

Line 355 TH17 cells cause inflammation and autoimmune disease, whereas TH1 cells are required for increased resistance to infections.

This is a strong sentence. Either soften it a little or specify ‘influenza A virus infections”, because Th17 cells/IL-17 have are involved in controlling other types of infections (e.g. Listeria sp., Streptococcus pneumoniae).

Line 369 T2DM decreased the overall amount of CD11c+ dendritic cells in subcutaneous adipose tissue, but increased the amounts of CD123+ plasmacytoid dendritic cells, contributing to the dysregulation of the lymphocyte immune response [88]

The cited paper used flow cytometry to show a percentage of cells, but did not backcalculate to cell number to report an absolute number of each cell population. Therefore, it’s not possible to tell whether (for example) a lower % CD11c+ DC meant that there were fewer CD11c+ DCs in the AT or a higher number of non-CD11c+ HLA-DR+ cells.

Line 451 In combination with high leptin levels, low adiponectin levels contribute to the pathogenesis of MetS and increased difficulty clearing infections caused by influenza virus and other pathogens.

To back up this conclusion, it would be beneficial if the authors cite primary papers that directly look at the effect of adiponectin on pathogen burden, in addition to the papers on adiponectin impact on inflammation and metabolic health already included from which we can infer there may be an impact on pathogen clearance.

Line 461 Additionally, low adiponectin levels occur in pregnant women and individuals with MetS [105], providing evidence that low adiponectin levels may contribute to increased susceptibility and advanced disease severity in multiple high-risk populations.

“Providing evidence” is a little strong for this reference/sentence. Perhaps taken together with the previous papers, it may suggest a contribution of adiponectin, but in isolation low adiponectin levels in high risk individuals is only correlative.

Figure 1 ‘induces’ and ‘leads’ should be capitalized

Author Response

Broad comments:

The "Cell-Mediated Responses" section reads as a general and light overview of the CMR in general, and more information or descriptions of studies specific to CD4 and CD8 responses during influenza or IAV immunization would improve this section, particularly as a backdrop to when the authors detail CMRs in the obese environment later in the manuscript.

  • Thank you for the comment. In response, this section has been revised and enhanced.

 Specific comments:

Line 202 According to one meta-analysis, metabolically healthy, obese individuals have a higher risk of mortality than do healthy weight individuals. Additionally, metabolically unhealthy groups in all BMI categories have a similarly elevated risk as do their metabolically healthy counterparts [46]. This suggests that metabolic health may be a stronger predictor of health than weight or BMI alone.

It would be helpful to reword this section, because the correlations seem to indicate that metabolic health is not a predictor of mortality, which is contrary to the authors’ conclusions.

  •  Corrected and revised for clarity. Thank you.

Line 253 The macrophages in obese patients can enter the AT and convert from an anti-inflammatory M2 alternative state to a proinflammatory M1 classical state to promote insulin resistance and T2DM [68]

Word missing between ‘convert’ and ‘from’

  • Corrected. Thank you.

Line 276 Indeed, the terms “adipaging” and “inflammaging” were coined to highlight how the chronic inflammation and dysfunctional AT in obese patients exhibits similar effects as aging

Missing a reference for inflamm-aging (Franceschi et al, 2000).

  • The term inflammaging was removed.

Line 444 Moreover, overly active NF-KB can alter innate responses and inflammation within the endothelium, providing a link between metabolic disease and premature aging that could impact premature aging [102]

This sentence needs to be made clearer or more in detail. Many factors activate NFkB altering inflammation, most of which are not related to premature aging. If the authors could explain in more detail why NFkB activation in this circumstance links metabolic disease and premature aging, that would strengthen the conclusion.

  • Revised as suggested.

Line 355 TH17 cells cause inflammation and autoimmune disease, whereas TH1 cells are required for increased resistance to infections.

This is a strong sentence. Either soften it a little or specify ‘influenza A virus infections”, because Th17 cells/IL-17 have are involved in controlling other types of infections (e.g. Listeria sp., Streptococcus pneumoniae).

  •  Revised as suggested.

Line 369 T2DM decreased the overall amount of CD11c+ dendritic cells in subcutaneous adipose tissue, but increased the amounts of CD123+ plasmacytoid dendritic cells, contributing to the dysregulation of the lymphocyte immune response [88]

The cited paper used flow cytometry to show a percentage of cells, but did not backcalculate to cell number to report an absolute number of each cell population. Therefore, it’s not possible to tell whether (for example) a lower % CD11c+ DC meant that there were fewer CD11c+ DCs in the AT or a higher number of non-CD11c+ HLA-DR+ cells.

  • Revised as suggested.

Line 451 In combination with high leptin levels, low adiponectin levels contribute to the pathogenesis of MetS and increased difficulty clearing infections caused by influenza virus and other pathogens.

To back up this conclusion, it would be beneficial if the authors cite primary papers that directly look at the effect of adiponectin on pathogen burden, in addition to the papers on adiponectin impact on inflammation and metabolic health already included from which we can infer there may be an impact on pathogen clearance.

  •  This section has been revised.

Line 461 Additionally, low adiponectin levels occur in pregnant women and individuals with MetS [105], providing evidence that low adiponectin levels may contribute to increased susceptibility and advanced disease severity in multiple high-risk populations.

“Providing evidence” is a little strong for this reference/sentence. Perhaps taken together with the previous papers, it may suggest a contribution of adiponectin, but in isolation low adiponectin levels in high risk individuals is only correlative.

  •  This section has been revised.

Figure 1 ‘induces’ and ‘leads’ should be capitalized

  • Correction made. Thank you

Reviewer 2 Report

Thoroughly processed paper is addressing very crucial point: highly important target groups in considerable risk in contrast with waning immunity and decreased vaccine efficacy in respective upper age cohort in combination with co-morbidities. 

Major comments:

  1. Authors citing the most recent flu season, however in many countries wearing protective masks decreased burden of flu essentially during the last season. COVID countermeasures influenced also dramatically flu. However in in countries with lower flu vaccine coverage it may increase the risk of more severe course of flu in following seasons, because of recent lower exposure. How to cope with this phenomenon?
  2. How to design future vaccines for elderly to tackle the issue of obesity?
  3. Can You explain relatively low efficacy of flu vaccines in comparison to COVID-19 vaccines?
  4. Do You suppose gender specific vaccines against flu?

Author Response

Thoroughly processed paper is addressing very crucial point: highly important target groups in considerable risk in contrast with waning immunity and decreased vaccine efficacy in respective upper age cohort in combination with co-morbidities. 

Major comments:

  1. Authors citing the most recent flu season, however in many countries wearing protective masks decreased burden of flu essentially during the last season. COVID countermeasures influenced also dramatically flu. However in in countries with lower flu vaccine coverage it may increase the risk of more severe course of flu in following seasons, because of recent lower exposure. How to cope with this phenomenon?
  • A section is now included commenting on this.
  1. How to design future vaccines for elderly to tackle the issue of obesity?
  • Great question. While we note that better vaccines are needed for vulnerable populations, in depth discussion on how to design such vaccines is outside the scope of this manuscript.
  1. Can You explain relatively low efficacy of flu vaccines in comparison to COVID-19 vaccines?
  • Great question. This is now discussed.
  1. Do You suppose gender specific vaccines against flu?
  • Interesting concept. While we note that there are some gender-specific differences in responses, more studies are needed to determine if these are large enough to be considered in vaccine design.